# A facial expression recognition network using hybrid feature extraction

**Dandan Song**ⓘ*, **Chao Liu**

Xinjiang Institute of Technology, Aksu, China

* calmskydan@sina.com

## Abstract

Facial expression recognition faces great challenges due to factors such as face similarity, image quality, and age variation. Although various existing end-to-end Convolutional Neural Network (CNN) architectures have achieved good classification results in facial expression recognition tasks, these network architectures share a common drawback that the convolutional kernel can only compute the correlation between elements of a localized region when extracting expression features from an image. This leads to difficulties for the network to explore the relationship between all the elements that make up a complete expression. In response to this issue, this article proposes a facial expression recognition network called HFE-Net. In order to capture the subtle changes of expression features and the whole facial expression information at the same time, HFE-Net proposed a Hybrid Feature Extraction Block. Specifically, Hybrid Feature Extraction Block consists of parallel Feature Fusion Device and Multi-head Self-attention. Among them, Feature Fusion Device not only extracts the local information in expression features, but also measures the correlation between distant elements in expression features, which helps the network to focus more on the target region while realizing the information interaction between distant features. And Multi-head Self-attention can calculate the correlation between the overall elements in the feature map, which helps the network to extract the overall information of the expression features. We conducted a lot of experiments on four publicly available facial expression datasets and verified that the Hybrid Feature Extraction Block constructed in this paper can improve the network's recognition ability for facial expressions.

## Introduction

Facial expression is a non-verbal communication that conveys a person's emotional state and intention. Facial expression recognition involves biology, sociology, psychology and other disciplines, as well as emotional computing, computer vision, pattern recognition, image processing and other fields, which has extremely high academic research significance and challenges. At the same time, facial expression recognition is widely used in many fields of society and has great practical significance, which can effectively improve the level of social governance and public services and improve the quality of human life. With the development of the artificial

**Data Availability Statement:** The experimental data supporting the results of this study are Facial Expression Recognition 2013, Face Expression Recognition Plus, RAF-DB dataset and Affectnet dataset. They can be obtained from the following websites: Facial Expression Recognition 2013:

https://www.kaggle.com/c/challenges-in-representation-learning-facial-expression-recognition-challenge/data Face Expression Recognition Plus: https://github.com/Microsoft/FERPlus RAF-DB dataset: http://www.whdeng.cn/RAF/model1.html Affectnet dataset: http://mohammadmahoor.com/affectnet/.

**Funding:** This research was funded by Natural Science Foundation of Xinjiang Uygur 474 Autonomous Region Grant2022D01C461.

**Competing interests:** The authors have declared that no competing interests exist.

intelligence field, research on facial expression recognition has become a hot topic in computer vision [1]. Therefore, it is crucial to construct a method with higher recognition accuracy and faster speed.

Early datasets for face expression recognition were usually obtained by researchers or professional actors filming under laboratory conditions. As the later training data transitioned from experimental to natural environments and the amount of data was relatively sufficient, many researchers began to shift the focus of their research from traditional methods using handcrafted features or shallow learning, such as Local Binary Patterns (LBP) [2], to deep learning methods.

With faster processing power and better network architecture, deep learning methods have achieved higher recognition accuracy in face expression recognition tasks. In the deep learning method, the convolutional neural network(CNN) [3] is a very representative image classification model. CNN uses convolution kernels of different sizes to extract the local semantic information in images and analyze the potential logical relationship of pixels, which can effectively identify the differences between different types of images and make correct classification. In addition, CNN-based models such as LeNet [4], VGG [5], GoogleNet [6], EigenFace [7], FisherFace [8], Siamese Network [9], Capsule Network [10], Face Attention Network [11], ResNet [12] and DenseNet [13] have achieved excellent classification performance in face recognition tasks. However, facial expression information is often unevenly distributed in different areas of the image, it is difficult for the pre-existing classification model to measure the relationship between pixels that make up a complete expression. This is because once the convolution kernel size is set, it can only extract the feature information in a limited area of the image, which makes it difficult for convolution neural network to effectively extract the complete expression features at one time.

Recently, the Transformer [14] model in the field of natural language processing has been introduced into the visual field, such as Vision Transformer (ViT) [15], Swin Transformer [16], etc. Due to their powerful global modeling ability, they have achieved good results in image classification tasks. Inspired by this, some scholars have combined Transformer with CNN, such as CvT [17]. Although models based on the fusion of the two can have both local information extraction and global modeling capabilities, it also brings the problem of excessive network computation. Some scholars use pure Transformer networks, such as TransFER [18] and ViL-BERT [19], but local information is also crucial for facial expression recognition tasks. Some scholars have also used the self-attention mechanism [20] in Transformer in facial recognition tasks, such as Residual Attention Network (RAN) [21], Cross-Modal Transformer (CMT) [22], DAMF-Net [23], DCT [24], TSPN [25], etc. It can be seen that the self-attention mechanism in Transformer has powerful global modeling ability by calculating the correlation between global pixels in the image. Although the complete expression features in the image can be extracted by using Transformer, the background information is introduced into the calculation of the logical relationship between pixels by Transformer, which often makes it difficult for the classification model to focus the learning center on the important expression feature areas.

Although convolution kernel can extract local details from images, its global modeling ability is insufficient. In addition, although the self-attention mechanism in Transformer has strong global modeling ability, it is easily disturbed by background information. Aiming at the deficiency of convolution kernel and the self-attention mechanism in facial expression feature extraction, we designed a facial expression recognition network called HFE-Net. In order to extract local information and global information from images synchronously, HFE-Net adopts the combination of convolution kernel and the self-attention mechanism [26]. In addition, HFE-Net also introduces a module Feature Fusion Device with different modeling methods.

The Feature Fusion Device can help the network to model facial features from multiple angles by calculating the correlation between distant pixels, so as to improve the recognition accuracy. In summary, our contributions are as follows:

(1) Aiming at the shortcomings of local modeling of convolution kernel and global modeling of Transformer, HFE-Net constructs a hybrid network of convolution kernel and Transformer to extract face features from different angles.

(2) The hybrid feature extraction module of Multi-head Self-attention mechanism and Feature Fusion Device improves the network's concentration on expression features.

(3) Extensive experimental results illustrate that our proposed method outperforms the state-of-art methods on different datasets.

## Related work

Facial Expression Recognition (FER) [27] is one of the important research directions in the field of computer vision and pattern recognition. With the development of deep learning technology, methods based on Convolutional Neural Network (CNN) and Transformer are one of the new research directions in the field of FER. With the continuous emergence of FER methods based on CNN and Transformer, the accuracy of FER has been greatly improved.

### Methods based on CNN

Since CNN was proposed, it has been absolutely dominant in the field of image classification. The application of CNN in the direction of FER began with the release of the FER2013 dataset in 2013. Since then, CNN have become one of the most commonly used deep learning models in the FER domain.

Mollahosseini et al. [28] proposed a CNN model based on VGGNet, which can shorten the training time of the model and achieve higher accuracy of expression classification than previous methods; Hossain et al. [29] proposed a facial expression recognition method based on depth residual network, which uses residual learning unit to solve the contradiction between network depth and model convergence, and can improve the accuracy of expression recognition; Lopes et al. [30] combined a simple convolutional neural network with specific image preprocessing technology to improve the accuracy of expression classification; Jung et al. [31] used two different convolutional neural network models to extract expression features, one of which extracted temporal appearance features from image sequences, and the other extracted temporal geometric features from temporal facial landmarks, and fused the features extracted by the two convolutional neural network models by adopting a new fusion method; Liu et al. proposed an Au-inspired deep networks [32] inspired by facial action units, which decomposed facial expressions into multiple facial action units, and then constructed convolutional neural networks to extract expression features from each facial action unit; FaceNet [33] is a CNN model mainly used for face recognition. This model adopts triplet loss function, which can train highly differentiated face features on a small training set and achieve very good recognition results. HoloNet [34] is a CNN model specially designed for FER. It uses CReLU and residual structure to increase the depth of the network, and uses the inception-residual block to learn multi-scale and expression discrimination features. In order to make full use of the complementary information between face and background features, Zhou et al. [35] proposed cross-attention (CA) block, element recalibration (ER) block and adaptive-attention (AA) block. Among them, cross-attention (CA) block is used to study and capture the

complementarity between facial expression features and their backgrounds. Element recalibration (ER) block modifies the feature map of each channel by embedding global information. Adaptive-attention (AA) block is used to infer the best feature fusion weight, and the adaptive emotional features are obtained through mixed feature weighting operation.

## Methods based on transformer

With the emergence of Transformer, many scholars have introduced it into image tasks. The first transformer structure for vision, ViT, was proposed by Dosovitskiy et al. for image classification and achieved impressive performance while bringing breakthroughs in the field of computer vision. In recent years, several researchers have begun to apply Transformer in the direction of FER. SCN [36] is a self-repairing network that effectively suppresses uncertainty and prevents deep learning from overfitting in facial expression recognition. SCN consists of three key modules: a self-attention mechanism, a ranking regularization, and a labeling correction mechanism, which can efficiently deal with both synthetic and real-world uncertainty and achieves results in public benchmarks that outperform current state-of-the-art methods. APViT [18] is the first work to use Transformer for face expression recognition, while a feature selection module is designed for fusing RGB and LBP inputs, and achieves good performance in the FER task. Two different types of attention pooling modules are used in this, attention patch pooling and attention marker pooling. These modules help the model to focus on the most discriminative features and ignore irrelevant information such as noise or occlusion, thus improving the accuracy of FER. Xue et al. [37] randomly deleted the self-attention module by using the Multi-head Self-attention discarding algorithm (MSAD), which made their model learning different from the rich relationship between local blocks. Zhao et al. [38] first used the convolutional spatial transformer guide the network to learn occlusion and pose-robust facial features from the spatial perspective. Temporary Transformer allows the network to learn contextual features from the temporary perspective. However, these models based on transformer simply divide a face picture into fixed-size patch when processing the input image, resulting in the loss of some key information between adjacent patch.

Aiming at the shortcomings of CNN and Transformer in facial expression recognition, Jiang et al. [39] proposed a novel graph-based model called Face2Nodes. Face2Nodes consists of two key parts: multi-scale feature fusion-based patch embedding and relationship-aware dynamic graph convolution network. Multi-scale feature fusion-based patch embedding uses multi-scale feature fusion mechanism to obtain more differentiated graph node features. The relationship-aware dynamic graph convolution network learns the potential information correlation between different nodes in the graph. In order to alleviate that CNN-based methods can't capture the detailed and key features that distinguish different facial expressions from a global perspective, Wu et al. [40] proposed a novel cross-hierarchy contrast (CHC) framework called FER-CHC. Specifically, FER-CHC uses CHC to adjust the feature learning of the backbone network and enhance the global representation of facial expressions. For the task of facial expression recognition, Huang et al. [41] proposed two attention mechanisms based on CNN. In low-level feature learning, the grid-wise attention mechanism captures the dependence of different regions from facial expression images. In advanced semantic representation, Visual Transformer Attention Mechanism uses a series of visual semantic tags to learn global representation.

Inspired by the aforementioned techniques, we attempted to explore a new network architecture that can simultaneously extract local and global information. In order to solve the above problems, we use CNN and Transformer in series, which makes the network have both CNN's local modeling ability and Transformer's global modeling ability. At the same time, we

use Feature Fusion Device and transformer in parallel to further extract the key information between adjacent patch.

## Methods

### Network architecture

The network we propose is a hybrid network that includes CNN and a Hybrid Feature Extraction Block. The Hybrid Feature Extraction Module includes a Feature Fusion Device and a Multi-head Self-attention(MHSA). The overall architecture diagram is shown in Fig 1. Firstly, the network obtains facial feature maps with rich local information through CNN. Then, the feature maps are sent to the Hybrid Feature Extraction Block, and the information interaction between adjacent patches is promoted through Feature Fusion Device, and the global features of faces are obtained through MHSA. After merging the two, the complementarity between different features can be effectively improved. Finally, facial expression recognition is carried out through the fused features.

EfficientNetV2 [42] not only has high accuracy, but also faster training speed with fewer parameters. Therefore, in CNN stage, we use EfficientNetV2 as the backbone network for feature extraction. The specific operation is shown in Fig 2, which contains 8 Stages in total. The feature map size of our input network is $X \in R^{3 \times 224 \times 224}$, after Stage0, the feature map size has increased to $24 \times 112 \times 112$, after 3 Fused MBConv feature maps, the size has been reduced to $64 \times 28 \times 28$. After 3 MBConv feature maps, the size has been reduced to $256 \times 7 \times 7$.

We first use three Fused-MBConv layers with small expansion rate, and then use three MBConv layers. Among them, Conv1, Conv4 and Conv6 represent convolution kernels with expansion rates of 1, 4 and 6 respectively. In the MBConv structure shown in Fig 2, firstly, we use a convolution kernel with the size of $1 \times 1$ to improve the channel feature dimension of the feature map. Then the features are extracted by using the Depthwise Convolution(DwConv) with the size of $3 \times 3$ and the SE module, which can effectively simplify the network parameters. Finally, the channel dimension is reduced by $1 \times 1$ convolution. MBConv can effectively avoid feature loss caused by compression and form a multi-range receiving domain of spatial

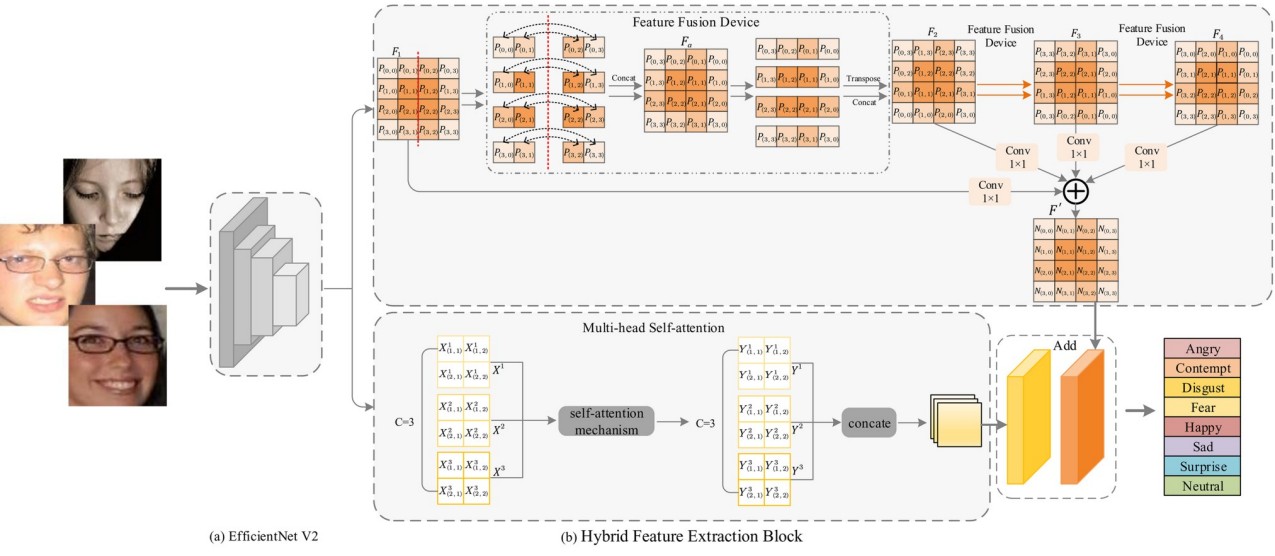

**Fig 1. The overall network architecture of HFE-Net.**

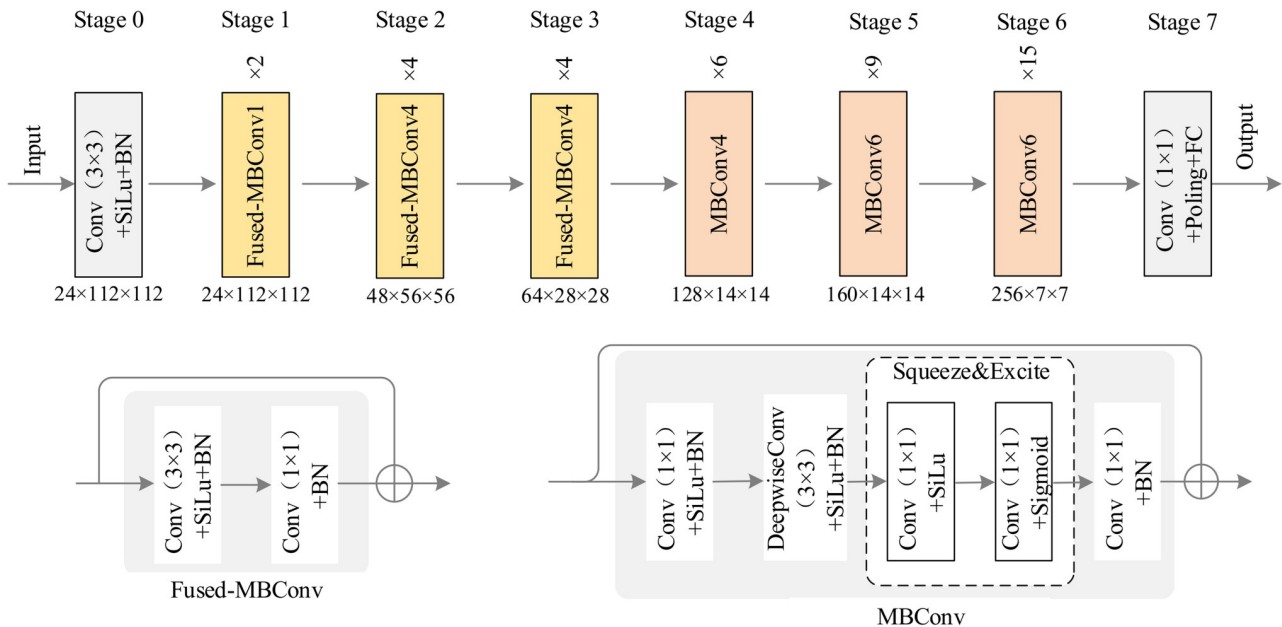

**Fig 2. EfficientNetV2 [42] network architecture.**

and channel positions through different convolution kernels to extract local details in the feature map. However, using DwConv in the shallow layers of the network will slow down the training speed, so the first three layers of the network are replaced with Fused-MBConV.

To further explore the detailed information in the feature map and perform long-range modeling, we adopt the structure of the Feature Fusion Device in parallel with Multi-head Self-attention for multi-angle feature extraction and fusion to achieve the exploration of positional relationships between elements in different ranges. Next, we will introduce the two parts of the Feature Fusion Device and Multi-head Self-attention in detail.

## Feature fusion device

In order to explore the relationship between the elements in the feature map and the elements that are far away from it on the basis of extracting more local details, we use Feature Fusion Device as shown in Fig 1, we demonstrate its workflow. Assume that the feature map $F_1 \in R^{1 \times 4 \times 4}$ passed into Feature Fusion Device, where 1 is the channel and $4 \times 4$ are the height and width of the feature map. Based on $F_1$, we transform the feature map according to a certain pattern to get $F_2$, $F_3$, $F_4$. Then we use $1 \times 1$ convolution to get the information interaction between the channels respectively, and finally get the feature map $F'$. It is worth noting that each element in $F'$ is obtained from the interaction of local region elements in the feature map $F_1$. The specific calculation steps are as in Eq (1):

$$N_{(i,j)} = \omega_1 P_{(i,j)} + \omega_2 P_{(j,H-1-i)} + \omega_3 P_{(H-1-i,W-1-j)} + \omega_4 P_{(W-1-j,i)} \tag{1}$$

Among them, $0 \geq i, j \leq 3$, $\omega_1, \omega_2, \omega_3, \omega_4$ is the parameter, $N_{(i, j)}$ are all composed of $F_1$, $F_2$, $F_3$, $F_4$ and parameters $\omega$ obtained by multiplying.

As shown in Fig 1, we demonstrate the specific transformation process of the feature map. Taking the original feature map $F_1$ as the basic unit, we divide the feature map into two parts along the column dimension. Next, the pixels in the two feature sequence are shifted in space.

Then, the two feature sequences are spliced to form a new feature map $F_a$. Finally, transform the positions of rows and columns according to a specific rule on the basis of $F_a$ to get $F_2$. Similarly, we transform the positions of $F_2$ and $F_3$ as the base unit to get $F_3$ and $F_4$, respectively.

From the schematic diagram of Feature Fusion Device, we can see that Feature Fusion Device not only further improves the extraction of local details, but also explores the relationship between elements that are far away from each other, for example, in $F_1$ the four elements at positions (0, 0), (0, 3), (3, 0), and (3, 3) are far away from each other, and it is very difficult to explore the relationship between them if they are simply passed into the feature map $F_1$. However, Feature Fusion Device's operation of transforming the feature map and multiplying it with the newly generated 3 feature maps by the parameters $\omega_1, \omega_2, \omega_3, \omega_4$ respectively to get the feature map $F_t$ makes $F_t$ contain the relationship between them, which is conducive to the network to better distinguish between different categories of elements and thus improve the performance of the network.

## Multi-head self-attention

As shown in Fig 3, we show the working principles of self-attention mechanism and Multi-head Self-attention mechanism respectively. In essence, the Multi-head Self-attention mechanism first divides the feature map into several groups in the channel dimension. Then, the global pixel correlation of different groups of feature maps is calculated by using the self-attention mechanism. Finally, the feature maps of different groups are combined along the channel dimension. Next, we will explain the specific workflow of Multi-head Self-attention in detail.

As shown in Fig 1, we demonstrate the workflow of Multi-head Self-attention. We assume that the feature map $X \in R^{3 \times 2 \times 2}$, where 3 is the channel and $2 \times 2$ is the height×width of the feature map. Next, feature map $X$ is divided along the channel dimension and feature maps $X^1 \in R^{1 \times 2 \times 2}$, $X^2 \in R^{1 \times 2 \times 2}$ and $X^3 \in R^{1 \times 2 \times 2}$ are obtained. Subsequently, feature maps $X^1$, $X^2$ and $X^3$ are respectively input to the self-attention mechanism module for global pixel correlation calculation and output feature maps $Y^1 \in R^{1 \times 2 \times 2}$, $Y^2 \in R^{1 \times 2 \times 2}$ and $Y^3 \in R^{1 \times 2 \times 2}$. Because their calculation flow is the same, we will explain the acquisition process of feature map $Y^1$ in detail.

Firstly, the spatial positions of pixels in the feature map $X^1$ are transformed to obtain image sequences $q \in R^{1 \times 4 \times 1}$, $k \in R^{1 \times 1 \times 4}$ and $v \in R^{1 \times 4 \times 1}$ respectively.

$$\{q, k, v\} = Reshape(\{X^1, X^1, X^1\}) \quad X^1 \in R^{1 \times 2 \times 2} \tag{2}$$

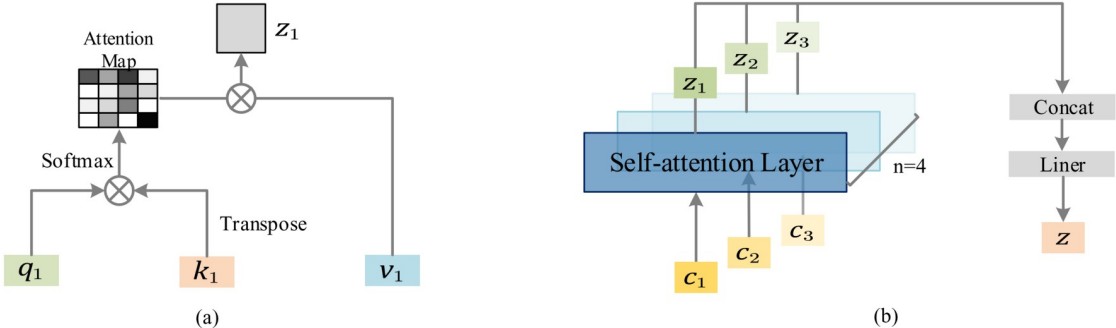

(a)　　　　　　　　　　　　　　　　　　　　　(b)

**Fig 3. (a) shows the working principle of self-attention mechanism, (b) shows the working principle of Multi-head Self-attention mechanism.**

Next, the image sequences $k = [q_1, q_2, q_3, q_4]^T$ and $q = \lfloor k_1, k_2, k_3, k_4 \rfloor$ perform matrix multiplication to get a new pixel sequence $\beta \in R^{1 \times 4 \times 4}$.

$$\beta = \begin{bmatrix} q_1 k_1 & q_1 k_2 & q_1 k_3 & q_1 k_4 \\ q_2 k_1 & q_2 k_2 & q_2 k_3 & q_2 k_4 \\ q_3 k_1 & q_3 k_2 & q_3 k_3 & q_3 k_4 \\ q_4 k_1 & q_4 k_2 & q_4 k_3 & q_4 k_4 \end{bmatrix} = MM(q, k) \quad \beta \in R^{1 \times 4 \times 4} \tag{3}$$

In Eq (3), $MM$ represents the dot product operation of pixels in two image sequences.

Next, we use the Softmax activation function to act on the pixel sequence $\beta$ to get the self-attention coefficient matrix $\omega \in R^{1 \times 4 \times 4} = \begin{bmatrix} \omega_{1,1} & \omega_{1,2} & \omega_{1,3} & \omega_{1,4} \\ \omega_{2,1} & \omega_{2,2} & \omega_{2,3} & \omega_{2,4} \\ \omega_{3,1} & \omega_{3,2} & \omega_{3,3} & \omega_{3,4} \\ \omega_{4,1} & \omega_{4,2} & \omega_{4,3} & \omega_{4,4} \end{bmatrix}$.

$$\omega_{i,j} = \text{Soft max}\left(q_i k_j\right) = \frac{e^{q_i k_j}}{\sum_{j=1}^{j=4} e^{q_i k_j}} \quad 1 \le i, j \le 4 \tag{4}$$

The self-attention coefficient and the pixels in the image sequence $v = [v_1, v_2, v_3, v_4]^T$ are dot-product, and the image sequence $Y^1 = [Y_1^1, Y_2^1, Y_3^1, Y_4^1]^T$ is obtained.

$$Y_i^1 = \sum_{j=1}^{j=4} \omega_{i,j} \times v_j \quad 1 \le i \le 4 \tag{5}$$

Finally, the pixels $Y_i^1 (1 \le i \le 4)$ in the image sequence $Y^1$ are arranged and combined to obtain the feature map $Y^1 = \begin{bmatrix} Y_{1,1}^1 & Y_{1,2}^1 \\ Y_{2,1}^1 & Y_{2,2}^1 \end{bmatrix}$.

Similarly, feature maps $X^2$ and $X^3$ output feature maps $Y^2$ and $Y^3$ through the self-attention mechanism module. The feature maps $Y^1$, $Y^2$ and $Y^3$ are spliced along the channel dimension to output a complete feature map $Y$.

$$Y = concate\{Y^1, Y^2, Y^3\} \quad Y^1, Y^2, Y^3 \in R^{1 \times 2 \times 2}; Y \in R^{3 \times 2 \times 2} \tag{6}$$

To sum up, the Multi-head Self-attention model allows to unite the information learned from different head parts, enabling the network to capture richer features.

## Loss function

The three datasets for facial expression recognition used in this paper all have a more serious category imbalance problem, which may lead to the model being too biased towards the majority category and not able to recognize the minority category well. The cross-entropy loss function can effectively alleviate this problem by weighting the loss of each category, so that the samples of the minority category get more attention in the loss calculation, thus prompting the model to better distinguish different categories. Therefore, cross-entropy loss is used as the loss function in this paper. The computational equation of CrossEntropyLoss is shown in Eq

(7):

$$\ell_{CrossEntropyLoss} = -\sum_{i=1}^{N} y(i) \bullet log(p(i)) \tag{7}$$

Among them, $N$ represents the number of expression categories, $y(i)$ represents the true label value of the image and $p(i)$ represents the predicted probability that the observed expression samples belong to expression category $i$.

## Experiment

### Datasets

Facial Expression Recognition 2013 (FER2013 dataset): This dataset contains approximately 32298 facial images with different expressions and consists of 48 × 48 pixel grayscale facial images. The face has been automatically registered, making it more or less centered and occupying approximately the same amount of space in each image. Its main labels can be divided into 7 types: anger, disgust, fear, happiness, sadness, surprise, and neutrality. We divided the FER2013 dataset into training and testing sets, and the specific data distribution is shown in Table 1.

Face Expression Recognition Plus (FERPlus dataset): This dataset was created by Emad Barsum et al. in 2016. The FERPLUS dataset contains 28259 grayscale images with a size of 48×48 pixels. These images cover 8 basic emotions: Anger, Content, Distust, Fear, Happy, Neutral, Sadness, Surprise. We divided the FERPLUS dataset into training and testing sets, and the data distribution for each category is shown in Table 2.

RAF-DB dataset: RAF-DB contains over 29670 facial images downloaded from the internet. The RAF-DB dataset selected 15339 images from seven basic expressions (i.e., Surprise, Wear, Distust, Happiness, Sadness, Anger, Neutral) for expression recognition. Each image is aligned and cropped, with a size of 100 × 100. The data distribution of each category in the RAF-DB dataset is shown in Table 3.

**Table 1. Distribution of data in the FER2013 dataset.**

| Dataset | Angry | Disgust | Fear | Happy | Sad | Surprise | Neutral | Total |
|---------|-------|---------|------|-------|-----|----------|---------|-------|
| Train | 3995 | 436 | 4097 | 7215 | 4965 | 4830 | 3171 | 28709 |
| Test | 491 | 55 | 528 | 879 | 626 | 594 | 416 | 3589 |
| Total | 4486 | 491 | 4625 | 8094 | 5591 | 5424 | 3587 | 32298 |

**Table 2. Distribution of data in the FERPlus dataset.**

| Dataset | Anger | Contempt | Disgust | Fear | Happy | Neutral | Sadness | Surprise |
|---------|-------|----------|---------|------|-------|---------|---------|----------|
| Train | 2100 | 119 | 119 | 532 | 7287 | 8740 | 3014 | 3149 |
| Test | 287 | 13 | 24 | 62 | 865 | 1182 | 351 | 415 |
| Total | 2387 | 132 | 143 | 594 | 8152 | 9922 | 3365 | 3564 |

**Table 3. Distribution of data in the RAF-DB dataset.**

| Dataset | Surprise | Fear | Disgust | Happiness | Sadness | Anger | Neutral | Total |
|---------|----------|------|---------|-----------|---------|-------|---------|-------|
| Train | 1290 | 281 | 717 | 4772 | 1982 | 705 | 2524 | 12271 |
| Test | 329 | 74 | 160 | 1185 | 478 | 162 | 680 | 3068 |
| Total | 1619 | 355 | 877 | 5957 | 2460 | 867 | 3204 | 15339 |

**Table 4. Distribution of data in the Affectnet dataset.**

| Dataset | Anger | Contempt | Disgust | Fear | Happy | Neutral | Sad | Surprise |
|---------|-------|----------|---------|------|-------|---------|-----|----------|
| Train | 24882 | 3748 | 3803 | 6377 | 134385 | 74863 | 25453 | 14086 |
| Test | 500 | 500 | 500 | 500 | 500 | 500 | 500 | 500 |
| Total | 25382 | 4248 | 4303 | 6877 | 134885 | 75363 | 25953 | 14586 |

Affectnet dataset: Affectnet contains 450,000 manually annotated facial expression images. In order to evaluate the model more fairly, we chose the basic facial expressions consistent with FERPlus to evaluate the accuracy. Among them, 287597 images are used as training sets and 4000 images are used as test sets. The data distribution of each category in the RAF-DB dataset is shown in Table 4.

## Implementation details

In all experiments in this article, the default image size for the input model is set to $224 \times 224$, batch size is set to 32, default to training the model for 100 epochs. In addition, we trained the model using the Adaw optimizer with an initial learning rate of 0.0001 and adjusted the learning rate using cosine annealing, where the hyperparameter T_max was set to 10. To ensure the fairness of the experiment, all experiments in this article share the same experimental environment and training parameters, using the same training and testing sets. Finally, all experiments in this article were trained and tested on a single NVIDIA TITAN 24G GPU.

## Evaluation indexs

It is necessary to choose an appropriate method to evaluate the recognition performance of the model. The output of the model designed in this paper is the prediction of facial expression, and the criterion for evaluating performance is the difference between the prediction graph and the label. The smaller the difference, the better the performance of the network. This article uses commonly used evaluation indicators: Accuracy, F1, Pre, Recall and Auc. The larger these evaluation indicators, the better the classification performance of the model.

## Ablation studies

As shown in Table 5, we conducted ablation experiments on the proposed model on four datasets, FER2013, FERPlus, RAF-DB and Affectnet to evaluate the effectiveness of the proposed modules and the impact of module composition on performance in more depth, where we named the models that used Multi-head Self-attention, Feature Fusion Device, multilayer perceptron(MLP), Multi-head Self-attention and Feature Fusion Device parallel structures as HFE-Net[1], HFE-Net[2], HFE-Net[3] and HFE-Net[4], respectively.

Impact of Multi-head Self-attention: As can be seen from Table 5, the recognition effect of HFE-Net[1] in four different FER datasets is better than that of baseline network. The experimental results show that Multi-head Self-attention can effectively assist the network to learn the complete expression features by calculating the correlation between the whole pixels in the feature map, which enhances the network's ability to recognize different expressions.

Impact of Feature Fusion Device: we also evaluated the performance of Feature Fusion Device in our ablation experiments. The results show that compared to baseline, the addition of the Feature Fusion Device module improves the accuracy on the four datasets by 1.34%, 0.85%, 1.27% and 0.75% respectively. The experimental results show that the large model can

**Table 5. Multilayer perceptron, Multi-head Self-attention and Feature Fusion Device ablation experiments on four FER datasets.**

| Datasets | Methods | Acc | F1 | Pre | Re | Auc |
|---|---|---|---|---|---|---|
| FER2013 | Effv2-s(baseline) | 69.91 | 67.52 | 66.82 | 68.74 | 81.81 |
| | HFE-Net[1] | 71.08 | 70.62 | 71.88 | 69.88 | 82.47 |
| | HFE-Net[2] | 71.25 | 70.99 | 72.85 | 69.74 | 82.41 |
| | HFE-Net[3] | 69.90 | 67.99 | 67.36 | 69.00 | 81.91 |
| | **HFE-Net[4]** | **71.69** | **71.32** | **72.32** | **70.60** | **82.88** |
| FERPlus | Effv2-s(baseline) | 87.71 | 79.23 | 81.03 | 77.99 | 87.99 |
| | HFE-Net[1] | 88.28 | 76.28 | 79.69 | 73.76 | 85.89 |
| | HFE-Net[2] | 88.56 | 77.27 | 86.85 | 73.32 | 85.71 |
| | HFE-Net[3] | 87.93 | 77.14 | 82.67 | 73.51 | 85.73 |
| | **HFE-Net[4]** | **88.72** | **78.21** | **87.43** | **73.58** | **85.85** |
| RAF-DB | Effv2-s(baseline) | 85.30 | 77.22 | 79.76 | 75.49 | 86.43 |
| | HFE-Net[1] | 86.15 | 79.25 | 81.46 | 77.75 | 87.64 |
| | HFE-Net[2] | 86.57 | 79.90 | 82.08 | 78.45 | 88.02 |
| | HFE-Net[3] | 84.29 | 76.42 | 80.08 | 74.38 | 85.77 |
| | **HFE-Net[4]** | **87.29** | **80.38** | **83.15** | **78.59** | **88.15** |
| Affectnet | Effv2-s(baseline) | 57.80 | 57.85 | 58.50 | 57.80 | 75.89 |
| | HFE-Net[1] | 58.28 | 58.35 | 58.88 | 58.28 | 76.16 |
| | HFE-Net[2] | 58.43 | 58.53 | 59.10 | 58.43 | 76.24 |
| | HFE-Net[3] | 58.33 | 58.28 | **60.05** | 58.33 | 76.19 |
| | **HFE-Net[4]** | **58.55** | **58.56** | 58.95 | **58.55** | **76.31** |

help the network focus its learning center on the target area by synchronously extracting local details from the feature map and measuring the correlation between distant pixels.

Impact of Multi-head Self-attention and Feature Fusion Device parallel structure: The results of the ablation experiments show that the use of a parallel structure(Feature Fusion Device and Multi-head Self-attention) in the model has better performance than a single-branch structure. It is worth mentioning that the model with the addition of Multi-head Self-attention and Feature Fusion Device parallel structure improves the accuracy by 1.78%, 1.01% and 1.99% compared to baseline. In parallel structure, Multi-head Self-attention can quickly obtain the complete information of expression features by calculating the correlation between global pixels. The large model not only extracts the local information in the expression features, but also measures the correlation between the distant elements in the expression features, which helps the network to focus more on the target area. Therefore, multi-angle features can be extracted by fusing different features.

In addition, we also carried out the related experiments of multilayer perceptron(MLP) in facial expression recognition. Experimental results show that HFE-Net[3] has better recognition ability in FERPlus and Affectnet than the baseline model, but its recognition performance in FER2013 and RAF-DB is worse than the baseline model. FERPlus and Affectnet have 8 kinds of pictures, while FER2013 and RAF-DB have 7 kinds of pictures. The experimental results show that MLP is helpful to improve the recognition performance of the network in the classification task with more categories.

## Comparison with state-of-the-art methods

In this section, we evaluate the proposed model on four datasets and compare the experimental results of the proposed model with other state-of-the-art models, respectively. As shown in

**Table 6. Comparison of experimental results on the FER2013 dataset.**

| Methods | Acc | F1 | Pre | Re | Auc |
|---|---|---|---|---|---|
| Resnet50 [12] | 69.46 | 68.63 | 71.97 | 66.62 | 80.70 |
| Efficientnet-B0 [43] | 69.85 | 69.20 | 71.42 | 67.94 | 81.38 |
| MobilenetV2 [44] | 68.01 | 66.71 | 67.63 | 66.25 | 80.39 |
| HireMLP-B [45] | 68.38 | 65.56 | 68.73 | 64.12 | 79.35 |
| CycleMLP-B4 [46] | 67.48 | 66.97 | 67.79 | 66.48 | 80.47 |
| Repmlp [47] | 68.82 | 67.00 | 66.99 | 67.29 | 80.98 |
| Repvgg-B1g2 [48] | 69.99 | 68.74 | 69.27 | 68.43 | 81.65 |
| Repvgg-B2g4 [48] | 69.55 | 69.31 | 70.10 | 68.70 | 81.75 |
| MoCoViT [49] | 67.26 | 66.01 | 67.48 | 65.54 | 79.97 |
| Biformer [50] | 69.21 | 68.00 | 68.79 | 67.45 | 81.10 |
| InceptionNeXt [51] | 66.70 | 64.30 | 64.39 | 64.51 | 79.42 |
| G-MixFormer [52] | 70.24 | 69.46 | 69.50 | 69.86 | 82.40 |
| Eff-CTM [53] | 68.26 | 67.40 | 67.51 | 67.78 | 81.19 |
| Eff-CTNet [54] | 69.32 | 69.10 | 70.21 | 68.34 | 81.54 |
| Effv2-s(baseline) | 69.91 | 67.52 | 66.82 | 68.74 | 81.81 |
| **HFE-Net[4]** | **71.69** | **71.32** | **72.32** | **70.60** | **82.88** |

Tables 6–9, our proposed model uses EfficientNetV2 as the backbone network for feature extraction, while the features are fused at the Hybrid Feature Extraction Block stage, and the loss function employs the cross-entropy loss.

In the comparison of advanced methods, Resnet50 has 49 convolutional layers and one fully connected layer. Due to the large number of layers in the Resnet50 network, it can better express features, thereby enhancing classification performance. EfficientNet-B0 is divided into 9 stages, and each stage containing multiple identical modules. Each module contains multiple MBConv structures and an SE module. By using composite coefficients to uniformly scale the

**Table 7. Comparison of experimental results on the FERPlus dataset.**

| Methods | Acc | F1 | Pre | Re | Auc |
|---|---|---|---|---|---|
| Resnet50 [12] | 87.21 | 77.25 | 81.84 | 74.30 | 86.11 |
| Efficientnet-B0 [43] | 87.96 | 79.45 | 86.24 | 75.57 | 86.78 |
| MobilenetV2 [44] | 86.96 | 75.89 | 79.88 | 73.00 | 85.41 |
| HireMLP-B [45] | 86.53 | 72.09 | 83.98 | 68.62 | 83.15 |
| CycleMLP-B4 [46] | 86.71 | 77.09 | 81.60 | 73.09 | 85.83 |
| Repmlp [47] | 87.62 | 76.16 | 85.42 | 72.30 | 85.13 |
| Repvgg-B1g2 [48] | 87.40 | 75.15 | 79.02 | 72.75 | 85.33 |
| Repvgg-B2g4 [48] | 87.09 | 77.06 | 81.35 | 74.50 | 86.26 |
| MoCoViT [49] | 85.71 | 74.28 | 83.59 | 69.85 | 83.71 |
| Biformer [50] | 87.81 | 75.14 | 81.61 | 71.46 | 84.71 |
| InceptionNeXt [51] | 85.96 | 75.24 | 84.33 | 71.40 | 84.53 |
| G-MixFormer [52] | 87.46 | 76.42 | 78.89 | 77.06 | 87.54 |
| Eff-CTM [53] | 87.06 | 76.46 | 79.34 | 74.31 | 86.08 |
| Eff-CTNet [54] | 87.75 | 75.65 | 81.95 | 72.21 | 85.09 |
| Effv2-s(baseline) | 87.71 | 79.23 | 81.03 | 77.99 | 87.99 |
| **HFE-Net[4]** | **88.72** | **78.21** | **87.43** | **73.58** | **85.85** |

**Table 8. Comparison of experimental results on the RAF-DB dataset.**

| Methods | Acc | F1 | Pre | Re | Auc |
|---|---|---|---|---|---|
| Resnet50 [12] | 84.29 | 76.96 | 80.07 | 74.65 | 85.87 |
| Efficientnet-B0 [43] | 84.32 | 76.79 | 76.86 | 76.95 | 87.08 |
| MobilenetV2 [44] | 81.94 | 74.08 | 77.20 | 71.97 | 84.34 |
| HireMLP-B [45] | 82.50 | 73.01 | 75.46 | 71.36 | 84.06 |
| CycleMLP-B4 [46] | 79.50 | 67.16 | 74.24 | 65.09 | 80.70 |
| Repmlp [47] | 84.52 | 77.10 | 79.11 | 75.68 | 86.43 |
| Repvgg-B1g2 [48] | 85.17 | 76.59 | 80.47 | 74.53 | 85.94 |
| Repvgg-B2g4 [48] | 84.65 | 77.73 | 80.12 | 75.97 | 86.59 |
| MoCoViT [49] | 79.60 | 69.75 | 72.29 | 68.02 | 82.15 |
| Biformer [50] | 84.32 | 76.09 | 76.11 | 76.21 | 86.74 |
| InceptionNeXt [51] | 81.10 | 72.10 | 71.67 | 72.74 | 84.70 |
| G-MixFormer [52] | 85.17 | 76.91 | 79.73 | 74.87 | 86.06 |
| Eff-CTM [53] | 83.54 | 75.29 | 77.20 | 74.39 | 85.71 |
| Eff-CTNet [54] | 84.91 | 78.16 | 82.46 | 75.61 | 86.41 |
| Effv2-s(baseline) | 85.30 | 77.22 | 79.76 | 75.49 | 86.43 |
| **HFE-Net[4]** | **87.29** | **80.38** | **83.15** | **78.59** | **88.15** |

three dimensions of depth, width, and input image resolution, thereby enabling the model to maintain high accuracy while maintaining high operational efficiency. EfficientNet V2 uses two modules, MBConv and Fused-MBConv, to extract feature map information; Hire MLP is based on channel aliasing MLP and rearrangement operation. By introducing internal region rearrangement and cross-region rearrangement, Hire MLP can aggregate global and local spatial information, so it has high flexibility and reasoning speed. RepMLP fully utilizes the advantages of convolution and MLP. The network captures local information by introducing convolution operations and incorporates convolution kernel weights into fully connected layers during the inference stage. In this way, the fully connected layer has local capture characteristics while maintaining global modeling capabilities. MoCoViT combines the advantages of CNN and Transformer, and its proposed mobile transformer module is carefully designed for mobile devices, very lightweight, and completed through two main modifications: the

**Table 9. Comparison of experimental results on the Affectnet dataset.**

| Methods | Acc | F1 | Pre | Re | Auc |
|---|---|---|---|---|---|
| Resnet50 [12] | 57.73 | 57.76 | 58.98 | 57.73 | 75.84 |
| Crossvit-15_224 [55] | 46.25 | 46.66 | 48.12 | 46.25 | 69.29 |
| Efficientnet-B0 [43] | 57.73 | 57.88 | 58.36 | 57.73 | 75.84 |
| MobilenetV2 [44] | 56.58 | 56.38 | 56.89 | 56.58 | 75.19 |
| HireMLP-B [45] | 54.23 | 54.44 | 55.86 | 54.22 | 73.84 |
| Repmlp [47] | 57.50 | 57.59 | 58.11 | 57.50 | 75.71 |
| Repvgg-B1g2 [48] | 57.55 | 57.30 | 57.90 | 57.55 | 75.74 |
| Repvgg-B2g4 [48] | 57.03 | 57.16 | 58.07 | 57.02 | 75.44 |
| MoCoViT [49] | 55.30 | 55.38 | 56.06 | 55.30 | 74.57 |
| Biformer [50] | 56.60 | 56.56 | 57.30 | 56.60 | 75.20 |
| Effv2-s(baseline) | 57.80 | 57.85 | 58.50 | 57.80 | 75.89 |
| **HFE-Net[4]** | **58.55** | **58.56** | **58.95** | **58.55** | **76.31** |

Mobile Self Attention (MoSA) module and the Mobile Feed Forward Network (MoFFN). MoSA simplifies the computation of attention maps through a branch-sharing scheme, while MoFFN serves as a mobile version of MLP in the transformer, further significantly reducing computational complexity. To solve the problems of high memory usage and high computational complexity, Biformer achieves more flexible computing power allocation through dual layer routing, allowing each query to process a small portion of the semantically most relevant K-V pairs, thereby achieving more flexible computing power allocation. The VGG series network, CycleMLP-B4, and MobilenetV2 have improved the model's ability to extract features by optimizing network architecture and improving network design. They have to some extent improved the accuracy in facial expression recognition tasks.

Although the above methods have achieved good results in the task of facial expression classification, they can't simultaneously extract important local features of facial expressions and explore the relationship between all pixels that make up a complete expression. On the contrary, HFE-Net4 proposed in this paper is a hybrid network architecture composed of convolution kernel and Transformer. It uses the powerful local modeling ability of convolution kernel and the powerful global modeling ability of Transformer to analyze the subtle changes of expressions in images and the relationship between pixels that constitute a complete expression. In addition, the Hybrid Feature Extraction Block proposed in this paper consists of Feature Fusion Device and Multi-head Self-attention Feature Fusion Device and Multi-head Self-attention extract multi-dimensional information from different angles, which is beneficial to enhance the performance of facial expression recognition in complex scenes. The experimental results show that our proposed model outperforms the existing state-of-the-art models on the FER2013, FERPlus, RAF-DB and Affectnet datasets. Among them, the accuracy reached 71.69%, 88.72%, 87.29% and 58.55% respectively, and F1 also reached the excellent level of 71.32%, 78.21%, 80.38% and 58.56% respectively. The experimental results demonstrate the effectiveness of our proposed method.

## Discussion

Figs 4 and 5 show the confusion matrices of Effv2-s (baseline) and our model on FER2013, FERPlus and RAF-DB respectively. FER2013, FERPlus and RAF-DB datasets not only have the problem of class imbalance, but also have very high similarity between different classes of

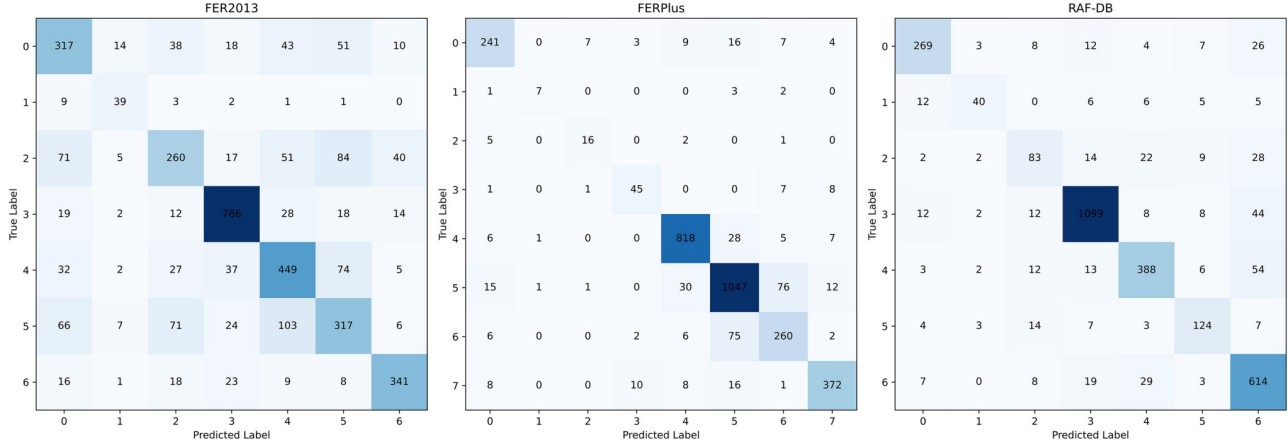

**Fig 4. Confusion matrix of Effv2-s (baseline) on three datasets.**

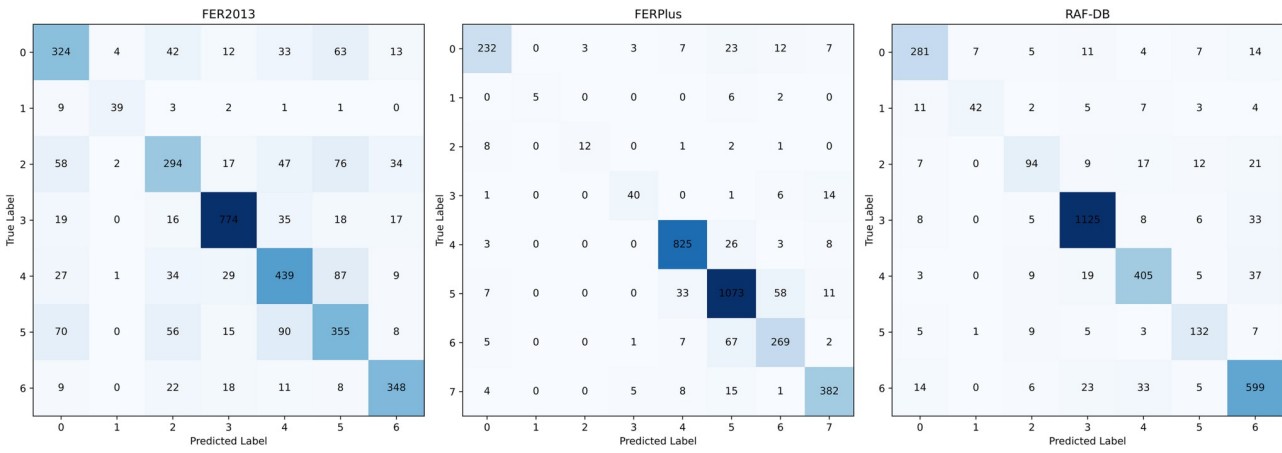

**Fig 5. Confusion matrix of the proposed model on three datasets.**

pictures. We observed that compared with the baseline model, the proposed model has a significant improvement in handling class imbalance and performance. This is because our model can not only use the advantages of convolution kernel and Transformer to extract the local information in the expression features in the image, but also measure the correlation between the pixels that make up the complete expression. In addition, the Hybrid Feature Extraction Block proposed in this paper can extract multi-dimensional features from different angles by using Feature Fusion Device and Multi-head Self-attention, which improves the model's ability to recognize expressions.

## Conclusion and prospects

In this article, we conducted in-depth research and exploration on the problems of facial expression recognition and proposed an effective solution. However, we also realize that future research work still needs further deepening and improvement. Therefore, the following section will focus on the main contributions of our research in future proposed work to demonstrate the academic value and practical significance of this paper.

Firstly, this paper provides a new theoretical perspective for research in the field of computer vision. Through a review and analysis of existing literature, we found that most studies have only focused on using CNN, Transformer, or fusing the two, which lack the ability to extract targeted features from specific task datasets. Based on this, we propose a novel network architecture HFE-Net that utilizes the CNN stage to extract shallow local features, and then uses the Hybrid Feature Extraction Block stage to extract global features and interact with multiple attention heads to construct semantic associations between different regions of interest, this enhances the network's ability to extract global features and effectively improves the accuracy of facial expression recognition tasks. This model can to some extent explain the shortcomings in existing research.

Firstly, this paper provides a new theoretical perspective for the research in the field of computer vision. By reviewing and analyzing the existing literature, we found that most of the research only focused on using CNN, Transformer or both, and lacked the ability to extract target features from specific task data sets. On this basis, we propose a new network architecture HFE-Net network. Because there are a large number of interference pixels in the original image that have nothing to do with expression, HFE-Net uses CNN to extract the expression information from the image region by region in the shallow stage of the network. Then,

HFE-Net uses the mixed feature extraction block to extract multi-dimensional features from different angles in the image, which effectively improves the accuracy of facial expression recognition tasks. This model can explain the shortcomings of the existing research to some extent.

Secondly, our research found that the effective use of multidimensional information from different angles in the feature map is an important factor to improve the network performance. To solve this problem, we use a Feature Fusion Device to move feature blocks in a specific pattern. Then, by calculating the relationship of pixels in the feature block, the Feature Fusion Device can simultaneously extract the local information in the expression feature and explore the correlation between distant elements in the expression feature. Finally, Multi-head Self-attention and Feature Fusion Device extract multi-dimensional features from different angles in parallel, which makes full use of the useful information in the feature map.

Again, our experimental results on four publicly available wild expression datasets demonstrate that our proposed model performs better than advanced methods. The research results of this paper also contribute to the promotion of the research perspective of fully utilizing feature map information in the field of computer vision to a certain extent. Our current research mainly focuses on discrete facial expression recognition tasks. To better apply the proposed model to real scenes, we will focus on continuous facial expression recognition research in the future. Through the publication of this paper, we hope that our ideas can arouse the interest of other scholars and encourage more researchers to engage in related research and learn from each other.

## Author Contributions

**Conceptualization:** Dandan Song.

**Formal analysis:** Chao Liu.

**Methodology:** Dandan Song.

**Software:** Chao Liu.

**Validation:** Dandan Song, Chao Liu.

**Writing – original draft:** Dandan Song.

**Writing – review & editing:** Dandan Song, Chao Liu.

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
