## [Decision Letter · Decision Letter 0]

1 Jul 2024

PONE-D-24-19404A Facial Expression Recognition Network Using Hybrid Feature ExtractionPLOS ONE

Dear Dr. song,

Thank you for submitting your manuscript to PLOS ONE. After careful consideration, we feel that it has merit but does not fully meet PLOS ONE’s publication criteria as it currently stands. Therefore, we invite you to submit a revised version of the manuscript that addresses the points raised during the review process.

**ACADEMIC EDITOR: **Thank you for submitting your manuscript to PLOSE ONE. We have received feedback from reviewers on your paper. While the reviewers acknowledge the potential of your work, they have raised several concerns that need to be addressed. In light of these comments, we invite you to submit a revised version of your manuscript that addresses the reviewers' concerns for further consideration. Please include a detailed response to each point raised by the reviewers, and clearly indicate the changes made in the manuscript.

We look forward to receiving your revised manuscript.

Kind regards,

Qionghao Huang

Academic Editor

PLOS ONE

Journal Requirements:

Reviewers' comments:

Reviewer's Responses to Questions

**Comments to the Author**

1. Is the manuscript technically sound, and do the data support the conclusions?

Reviewer #1: Partly

Reviewer #2: Yes

2. Has the statistical analysis been performed appropriately and rigorously? 

Reviewer #1: Yes

Reviewer #2: N/A

3. Have the authors made all data underlying the findings in their manuscript fully available?

Reviewer #1: Yes

Reviewer #2: Yes

4. Is the manuscript presented in an intelligible fashion and written in standard English?

Reviewer #1: Yes

Reviewer #2: Yes

5. Review Comments to the Author

Reviewer #1: The paper introduces a facial expression recognition network called HFE-Net, designed to capture both subtle changes in expression features and overall facial expression information. The method has been extensively tested on three public facial expression datasets, validating that the hybrid feature extraction block can enhance the network’s ability to recognize facial expressions. However, there are several areas that need improvement:

-In the Introduction section, the research problem posed by the authors is not very clear. Please clearly state the current Research Gap, how your work differs from existing studies, and briefly outline your technical contributions.

-In the Related Work section, it would be beneficial to include some recent related works (e.g., Face2nodes: learning facial expression representations with relation-aware dynamic graph convolution networks, INS, 2023; FER-CHC: Facial expression recognition with cross-hierarchy contrast, ASOC, 2023; Emotion recognition from large-scale video clips with cross-attention and hybrid feature weighting neural networks, ERPH, 2023; Facial expression recognition with grid-wise attention and visual transformer, INS,2021). This would provide readers with a more current understanding of this field.

-Regarding model design, the authors use two Feature Extraction Modules. The motivation behind this design is not very clear, and it’s uncertain whether other feature extraction networks could also be used. Please conduct some data analysis in the experimental section to confirm the rationality of the architecture design.

-In terms of experiments, it is recommended to add more datasets, such as Affectnet, to better demonstrate the model’s generalizability.

-It is suggested to add comparisons with the latest methods from recent years, as the baselines currently compared are not state-of-the-art (SOTA) methods.

-The paper still has some issues with language grammar or fluency. Please carefully check and revise.

Reviewer #2: This paper presents a facial expression recognition network by proposing a Hybrid Feature Extraction Block, which consists of parallel Big Model and Multi-head Self-attention. Overall, the paper is well presented. Nevertheless, I have a few concerns about the paper on the following points:

What are the authors' explanations for using Big Model rather than already existing methods in capturing long-range dependencies in feature maps?

The explanation of equation 2 states the Softmax is normalized, but the equation doesn’t have a normalization factor. Please correct this mismatch.

In the Cross-entropy loss function, equation 6, both variables are the same (p(i)), which is inaccurate because the loss is calculated between the predicted and true values.

The authors' discussion of why the proposed method outperformed existing approaches would provide more insights into their proposed network.

Labeling the confusion matrices with associated expressions rather than the indices can make them better for comparison.

6. PLOS authors have the option to publish the peer review history of their article (what does this mean?). If published, this will include your full peer review and any attached files.

Reviewer #1: No

Reviewer #2: No

---

## [Author Response · Author response to Decision Letter 0]

23 Aug 2024

Reviewer #1: The paper introduces a facial expression recognition network called HFE-Net, designed to capture both subtle changes in expression features and overall facial expression information. The method has been extensively tested on three public facial expression datasets, validating that the hybrid feature extraction block can enhance the network’s ability to recognize facial expressions. However, there are several areas that need improvement:

1. In the Introduction section, the research problem posed by the authors is not very clear. Please clearly state the current Research Gap, how your work differs from existing studies, and briefly outline your technical contributions.

Reply：Dear reviewer, as per your request, I have rephrased the issues with current facial expression classification models in the introduction section and highlighted the technical contributions of this paper. Specifically, the current facial expression recognition mainly adopts a network architecture based on convolutional neural networks and Transformers. Among them, convolutional kernels have strong local modeling capabilities but insufficient global modeling capabilities. Although Transformer can calculate the correlation of global pixels in an image, it also introduces too much background in the feature extraction process, which can lead to a decrease in the model's focusing ability. In response to the above issues, the HFE-Net proposed in this article uses convolutional kernels and multi head self attention mechanisms to extract features from different angles in images. In addition, this article also uses Big Model (Renamed as: Feature Fusion Device) with different modeling methods to calculate the correlation of distant elements to improve the network's ability to focus on facial expression features.

2.In the Related Work section, it would be beneficial to include some recent related works (e.g., Face2nodes: learning facial expression representations with relation-aware dynamic graph convolution networks, INS, 2023; FER-CHC: Facial expression recognition with cross-hierarchy contrast, ASOC, 2023; Emotion recognition from large-scale video clips with cross-attention and hybrid feature weighting neural networks, ERPH, 2023; Facial expression recognition with grid-wise attention and visual transformer, INS,2021). This would provide readers with a more current understanding of this field.

Reply：Dear reviewer, according to your recommendation, I have added the relevant introduction of predecessors' work in the relevant work section.

3.Regarding model design, the authors use two Feature Extraction Modules. The motivation behind this design is not very clear, and it’s uncertain whether other feature extraction networks could also be used. Please conduct some data analysis in the experimental section to confirm the rationality of the architecture design.

Reply：Dear reviewers, the motivation of using Multi-head self-attention mechanism and Big Model (Renamed as: Feature Fusion Device) in this paper comes from the insufficient local modeling ability of convolution kernel and the insufficient global modeling ability of Transformer. First of all, the model has global modeling ability, which can better extract the complete facial expression features. Secondly, the local modeling ability can better assist the network to focus the learning focus on the foreground information area in the image. For the validity of the argument put forward in this paper, I added the related experiment of Multilayer Perceptron(MLP) to the ablation experiment.

4. In terms of experiments, it is recommended to add more datasets, such as Affectnet, to better demonstrate the model’s generalizability.

Reply：Dear reviewers, according to your request, I have added related contrast experiments and ablation experiments in the Affectnet data set.

5. It is suggested to add comparisons with the latest methods from recent years, as the baselines currently compared are not state-of-the-art (SOTA) methods.

Reply：Dear reviewers, according to your request, I have increased the experimental results of the latest methods in recent years.

6. The paper still has some issues with language grammar or fluency. Please carefully check and revise.

Reply：Dear reviewer, according to your request, I have re-read the article and revised the relevant grammar.

Reviewer #2: This paper presents a facial expression recognition network by proposing a Hybrid Feature Extraction Block, which consists of parallel Big Model and Multi-head Self-attention. Overall, the paper is well presented. Nevertheless, I have a few concerns about the paper on the following points:

1. What are the authors' explanations for using Big Model rather than already existing methods in capturing long-range dependencies in feature maps?

Reply：Dear reviewer, I would like to make the following explanation for your question. The existing feature extraction modules are mainly composed of convolution kernel, Transformer, MLP and so on. Among them, the feature pyramid captures the long-term dependence in the feature graph by using convolution kernels of different sizes, but this method increases the computational complexity of the network. Similarly, although other feature extraction modules have realized the capture of remote context information, they often increase the computational burden of the network. Then, Big Model (Renamed as: Feature Fusion Device) realizes the capture of remote context information by shifting the spatial position of image pixels, which not only reduces the computational complexity of the network, but also improves the classification performance of the network.

2. The explanation of equation 2 states the Softmax is normalized, but the equation doesn’t have a normalization factor. Please correct this mismatch.

Reply：Dear reviewer, I have corrected this problem. Thank you for your guidance.

3. In the Cross-entropy loss function, equation 6, both variables are the same (p(i)), which is inaccurate because the loss is calculated between the predicted and true values.

Reply：Dear reviewer, I have corrected this problem. Thank you for your guidance.

4.The authors' discussion of why the proposed method outperformed existing approaches would provide more insights into their proposed network.

Reply：Dear reviewers, according to your request, we have explained why the proposed method is superior to the existing methods in Comparison with state-of-the-art methods, Discussion and Conclusion and prospects.

5.Labeling the confusion matrices with associated expressions rather than the indices can make them better for comparison.

Reply：Dear reviewers, this article provides our experimental results by imitating the relevant experiments provided in previous articles.

---

## [Decision Letter · Decision Letter 1]

7 Oct 2024

A Facial Expression Recognition Network Using Hybrid Feature Extraction

PONE-D-24-19404R1

Dear Dr. song,

We’re pleased to inform you that your manuscript has been judged scientifically suitable for publication and will be formally accepted for publication once it meets all outstanding technical requirements.

Kind regards,

Qionghao Huang

Academic Editor

PLOS ONE

Additional Editor Comments (optional):

Reviewers' comments:

Reviewer's Responses to Questions

**Comments to the Author**

1. If the authors have adequately addressed your comments raised in a previous round of review and you feel that this manuscript is now acceptable for publication, you may indicate that here to bypass the “Comments to the Author” section, enter your conflict of interest statement in the “Confidential to Editor” section, and submit your "Accept" recommendation.

Reviewer #1: All comments have been addressed

Reviewer #2: All comments have been addressed

2. Is the manuscript technically sound, and do the data support the conclusions?

Reviewer #1: Yes

Reviewer #2: Yes

3. Has the statistical analysis been performed appropriately and rigorously? 

Reviewer #1: Yes

Reviewer #2: Yes

4. Have the authors made all data underlying the findings in their manuscript fully available?

Reviewer #1: Yes

Reviewer #2: Yes

5. Is the manuscript presented in an intelligible fashion and written in standard English?

Reviewer #1: Yes

Reviewer #2: Yes

6. Review Comments to the Author

Reviewer #1: The author's response has addressed my concerns on this paper, and I recommend an acceptance of this paper.

Reviewer #2: The authors have addressed the reviewers' comments and revised the manuscript accordingly. Good luck.

7. PLOS authors have the option to publish the peer review history of their article (what does this mean?). If published, this will include your full peer review and any attached files.

Reviewer #1: No

Reviewer #2: No

---

## [Editor Report · Acceptance letter]

5 Nov 2024

PONE-D-24-19404R1 

PLOS ONE

Dear Dr. Song, 

I'm pleased to inform you that your manuscript has been deemed suitable for publication in PLOS ONE. Congratulations! Your manuscript is now being handed over to our production team.

Kind regards, 

on behalf of

Dr. Qionghao Huang 

Academic Editor

PLOS ONE